# Estimating the size of hard to sample populations: A comprehensive study on female sex workers and sexually exploited minors in Rwanda using privatized network sampling in 2023

Elysée Tuyishime[1*], Catherine Kayitesi[2], Eric Remera[3], Samuel Sewava Malamba[4], Ignace Habimana Kabano[1], Angela Unna Chukwu[1,5]

1 University of Rwanda–African Center of Excellence in Data Science (ACE-DS), Kigali, Rwanda, 2 HIV/AIDS and STIs Division, Institute of HIV Disease Prevention and Control, Rwanda Biomedical Centre (RBC), Kigali, Rwanda, 3 Research, Innovation and Data Science Division, Institute of HIV Disease Prevention and Control, Rwanda Biomedical Centre (RBC), Kigali, Rwanda, 4 Health Research and Evaluation Consultant, Kampala, Uganda, 5 Department of Statistics, University of Ibadan, Ibadan, Nigeria

* telisha.net@gmail.com

## Abstract

### Introduction

Female sex workers (FSW) are at increased risk of HIV and other STI. In addition, the burden of HIV infection among this group is much higher when compared to adult females in the general population. Estimating the number of FSW helps HIV/STI prevention through program design, planning, and implementation. The aims of this study are to provide the most up to date national population size estimates (PSE) and geographical distribution of female sex workers and sexually exploited minors in Rwanda. Having population size estimates of the HIV-mostly affected population, FSW in this case provides the basis for determining the denominators to assess HIV program performance towards national and global targets of controlling the HIV epidemic among the FSW population.

### Methods

Data were collected from May 8th to June 24th, 2023, across 10 study sites countywide. Privatized network sampling (PNS) was used, which is a population size estimation method that uses the network information collected within a bio-behavioral survey (BBS) that used respondent-driven sampling (RDS). To estimate the FSW and sexually exploited minors' population size, three PNS estimators were used: Cross-Sample, Cross-Alter, and Cross-Network.

**Data availability statement:** The data underlying the results presented in the study are available for researchers who meet set criteria upon request and approval by RBC (data access requests are sent at: 'info@rbc.gov.rw').

**Funding:** The author(s) received no specific funding for this work.

**Competing interests:** I have read the journal's policy and the authors of this manuscript have no competing interests.

## Results

The national-level FSW population size was estimated at 98,587 (95% CI: 82,978–114,196), corresponding to 2.3% of the total adult female population aged 15 years and above in Rwanda. We estimated that in the City of Kigali, 5.3%, in the West Province, 2.2%, in the East and South province, 1.7% each, and in the North province 1.6% of adult female population 15 years of age and older who were FSW.

## Conclusion

This was the first time that PNS was implemented as a PSE method in Rwanda, adding to the emerging tools that we have in the hard-to-reach PSE field. The PSE provides fundamental information to design, plan, and implement programs for FSW at the provincial level in Rwanda. Furthermore, these estimates will help to generate positive policy changes and to advocate for resources that will help in the effort to achieve a sustained HIV epidemic control in the country.

## Introduction

Globally, gay men and other men who have sex with men, sex workers, transgender people, people who inject drugs, and people in prisons and other closed settings are considered the five main key populations (KPs) that are particularly vulnerable to HIV and frequently lack adequate access to health services [1]. Some studies conducted on the HIV/AIDS epidemic have revealed the high burden of HIV infection among KPs [2].

The World Health Organization (WHO) highlights the need to focus on KP who are particularly vulnerable and disproportionately affected by HIV due to risk behaviors, marginalization, and structural factors such as stigma, discrimination, violence, human rights violations, and criminalization, which contribute to a lack of access to prevention and treatment services and hence become the key drivers of new HIV transmission in the general population [1,2]. During 2022, in Eastern and Southern Africa, 54% of the total number of new HIV infections were reported among the general population, and the remainder were reported among KPs, with 13% of those reported among female sex workers (FSW) [3].

In Rwanda, FSWs are considered among key population groups for HIV prevention and treatment focus due to their high-risk behaviors for contracting and/or transmitting STIs/HIV, mainly because of multiple partners and a low rate of condom use. They are often stigmatized and marginalized and relatively disproportionately affected by HIV. Due to a continued observed high prevalence of HIV among FSW compared to the general population, FSWs are considered a persistent niche of HIV and are thus treated as the bridge of HIV infection to the general population [4–6]. The HIV prevalence among FSWs in Rwanda has exhibited a downward trend over the years [7]. In 2010, the HIV prevalence among FSW in Rwanda stood at 50.8% [5]. Subsequently, in 2015, it decreased to 45.8% [8], and the most recent survey conducted in 2019 indicates a further reduction to 35.5% [9].

The landscape of sex work is undergoing rapid transformation with the rise of digital technologies, significantly altering how sex workers connect with clients and organize their work. Globally, increasing access to mobile phones, internet, and social media platforms has enabled a shift from venue-based to home-based and online sex work, offering greater autonomy but also introducing new challenges for outreach and surveillance [10,11]. In Rwanda, while empirical data are limited, anecdotal evidence and emerging trends suggest a similar shift, particularly among younger FSWs. This evolution makes traditional population size estimation (PSE) methods less effective, as they are typically reliant on visibility in public venues. Addressing this gap, there is a call for a well-suited method to capturing hidden subgroups such as digital and home-based FSWs and include them explicitly in both the sampling frame and analysis.

Sexually exploited minors represent a particularly vulnerable subgroup within the broader population of individuals engaged in sex work, yet they are often underrepresented in research due to ethical, legal, and methodological challenges [12,13]. Their involvement in commercial sex is typically driven by coercion, poverty, or exploitation, placing them at heightened risk of HIV infection and other adverse health outcomes [14]. Recognizing this, our study included sexually exploited minors, this refers to females aged 15–17 years engaged in sex work.

Estimating the size of the female sex worker (FSW) population is critical not only for informing program design, planning, and implementation but also for guiding evidence-based HIV prevention and treatment strategies. Accurate population size estimates (PSE) provide the necessary denominator to assess service coverage, identify gaps, and allocate resources effectively [15,16]. Moreover, PSE data support national and regional policy-making by helping stakeholders prioritize interventions, monitor progress toward public health goals, and ensure that high-risk and underserved populations, such as FSWs, are adequately reached and supported in the HIV response [17,18]. Surveying this group and obtaining reliable population estimates is challenging, as their marginalization often results in limited visibility within communities, making them a hard-to-reach population.

Various methods have been used thus far to estimate the population sizes of hard-to-reach groups, each with its own set of strengths and drawbacks. Respondent Driven Sampling (RDS) has been proven to be able to penetrate the networks of hard-to-reach populations, by leveraging peer networks where individuals are socially connected [19–21], however, this approach assumes random recruitment within well-connected social networks, yet this assumption is often violated among marginalized populations like FSWs, resulting in biased estimates [20,22]. Capture-recapture [23], while useful, require rigorous matching across data sources and can be compromised by mobility, alias use, or incomplete records, particularly in stigmatized populations [24]. Multiplier [25], though simple to implement, heavily relies on accurate service data and consistent population overlap between survey and service use, which may not always hold in practice [18]. The network scale-up method [26], and successive sampling [27] approaches are all common and provides an indirect estimation approach by asking respondents about the number of people they know in the target population. However, it is sensitive to transmission errors and assumes accurate knowledge and reporting of network ties, which can be difficult to validate [28]. Unfortunately, there is no gold standard approach, and various methodologies frequently result in contradictory conclusions [29].

Given these challenges, Privatized Network Sampling (PNS) presents a promising alternative. PNS is a networked capture-recapture approach that gains from the RDS and CRC strengths, and is based on the same foundation but collects additional important information for the estimation of the size of the hard-to-reach population [30,31]. PNS is particularly well-suited for hidden, stigmatized groups like FSWs and has demonstrated the ability to improve estimation accuracy while reducing social desirability bias and disclosure risk [32]. Thus, PNS addresses many limitations of traditional methods and offers a valuable innovation for population size estimation in HIV surveillance.

Privatized Network Sampling (PNS), population size estimation method was suggested for this study. This uses the network information collected within a bio-behavioral survey (BBS) that used RDS with further information collect on how individuals are networked to determine PSE.

In Rwanda, there have been three rounds of FSW PSE dated 2010 [33], 2018 [34]and 2022, all using Time Location Sampling (TLS). As the digital era emerges, with technologies reshaping and reorienting sex markets [35,36], the use of venue-based sampling approaches might be missing a chunk of FSW who never congregate in the venues, including those practicing sex work at home (home-based) and those who get clients from internet platforms (internet-based).

Repeated and accurate estimation of female sex worker (FSW) populations is essential for effective HIV prevention, policy planning, and resource allocation. In Rwanda, previous PSEs have often underestimated this population, particularly among internet-based and home-based FSWs. These gaps highlight the need for updated methods that reflect the evolving dynamics of sex work. This study addresses these shortcomings using Privatized Network Sampling (PNS) [37], capable to tap into previously unexplored FSW subgroups such as home and internet-based FSWs, offering more accurate and timely data to inform public health strategies.

## Materials and methods

### Study population

The PSE included peer-identified female sex workers (FSW), defined as any female assigned at birth aged 15 years and above whose primary source of income was commercial sex work (i.e., exchanging sex for money, goods, or services). Eligibility also required engagement in commercial sex work within the past 12 months, ability to communicate in one of Rwanda's official languages (English, French, Kinyarwanda), and provision of informed consent.

### Study design and procedures

To inform the development of this estimation, a formative assessment (FA) was conducted. A group meeting that included implementing partners (IP), stakeholders, and FSWs was convened, and focus group discussions (FGD) were conducted. FSWs from 5 provinces in Rwanda came to Kigali for a one-day meeting on March 10th, 2023. The objectives of the FA included the identification of sociocultural factors limiting or facilitating access to FSW, assessing the feasibility of the planned method and procedures, and identifying barriers and strategies to overcome them.

Representatives from implementing partner institutions serving FSW communities participated in the FA, as along with 10 purposively selected FSWs from five provinces. The selection aimed to ensure diversity in demographic characteristics and sex work modalities rather than representativeness, given the qualitative, given the qualitative nature of the assessment. Among the participating FSWs, 2 were under 20 years old, 2 were home-based, 1 uses internet-based platforms to reach clients, 2 were part of a university FSW network, and 3 engaged in street- or venue-based sex work. The FA focused on informing three key themes: study design and procedures, characteristics of the study population, and survey logistics.

Under the study design and procedures theme, we gained assurance in the appropriateness of the proposed method (PNS), and decided to use a combination of name initials and the last five digits of participant's phone number to create unique identification codes. The consultations also informed study site preparation and layout, guided the design of recruitment coupons, informed coupon (invitation) design, highlighted the importance of local government, and provided insights for developing training materials for data collectors. Regarding the second theme, we learned that some FSW subgroups are extremely hard to reach, including FSWs in the university students' networks, those using pimps, and middlemen, and this information guided us on the outreach strategies. Finally, the FA has informed the logistic component of the study, such as compensation for participation, including a FSW within each of the study teams to serve as a receptionist, and consideration for an electronic coupon (invitation).

This estimation utilized one single method of population size estimation, Privatized Network Sampling (PNS). This method utilized network data collected using the questions specifically developed for this purpose. Within a bio-behavioral survey (BBS) questionnaire that used RDS to sample FSW, more questions were added collecting

information on the degrees at which FSW population are networked, hence used for size estimation purposes. In respondent-driven sampling (RDS), initial participants, known as "seeds," are often purposively selected to initiate recruitment chains that can penetrate diverse segments of the target population. In our study, seeds were chosen to encompass a range of ages, geographic locations, and sex work modalities, aiming to capture the heterogeneity of the FSW community in Rwanda based on FA findings. This strategy is consistent with established RDS methodologies, which recognize that while seed selection is non-random, the subsequent peer recruitment process, if sufficiently deep and diverse, can mitigate initial selection biases and lead to a sample that reflects the broader population characteristics [38].Data were collected countywide in 10 study sites, which included Gihundwe Health Center (HC), Kibuye HC, Gisenyi HC in the West; Gitarama HC and Rango HC in the South; Muhoza in the North; Mukarange HC and Nyagatare HC in the East; Remera HC and WE-ACT FOR HOPE CLINIC in the City of Kigali. Data collection was performed between May 8th, 2023, and June 24th, 2023.

The PNS sampling followed the Respondent Driven Sampling (RDS), which is a probability-based chain-referral sampling methodology used to sample FSWs for a biobehavioral survey (BBS). For the RDS process, initial survey participants ("seeds") were purposively recruited by the survey team to start enrollment. Criteria to be a seed, one should have been engaged in commercial sex work at least 12 months prior to the estimation, well-connected within FSW social networks, well regarded by peers, able to communicate with data collectors, and supportive of estimation goals. Furthermore, three seeds by site were purposively recruited reflecting diversity in sociodemographic characteristics (e.g., age, sexual orientation and gender identity, education, area of residence, marital status, language, religion), HIV status, and affiliation with a KP organization or KP service provider.

Seeds and other subsequent recruiters were provided with a maximum of 3 coupons to distribute to their peers in their FSW social circle for recruitment. Instructions for peer recruitment using a recruitment process script were provided to seeds and participants by staff at the study site. When potential recruits came to the survey site, they were screened for eligibility and enrolled if they met the survey inclusion criteria and consented to participate; at this point, they were considered participants. After participating in the survey, these individuals were given their own recruitment coupons and asked to distribute them to their peers that they knew are FSW. This process continued until the target sample size and survey parameters were achieved.

However, RDS data contains limited information about participant's network. To collect major identifiable information about how individuals in the sample are related to one another, for each recruited respondent, using a cryptographic hash function, a hashed (anonymized) ID was created from the initials of the first and last name and the last 5 digits of the respondent's phone number using Tele funked coding [39].

To protect participant confidentiality while ensuring reliable tracking of recruitment patterns, we used a secure hashing technique to create unique identifiers from a combination of name initials and the last five digits of the participant's phone number, and this should only be spelt out by the respondent herself. This approach, adapted from methods described by Weinert [40], ensures that the identifiers are anonymized and non-reversible, aligning with best practices in data anonymization for hidden populations. Hashing was performed using a cryptographic algorithm, and the output was stored in a way that made re-identification impractical, thus preserving participant privacy while supporting the integrity of the recruitment process.

Furthermore, each recruited respondent was asked how many of the total network size peers the recruit knows their name and phone numbers, and a hashed ID was also created for up to 5 peers in the respondent's personal network. If the respondent stated knowing 5 or fewer peers, a hashed ID was created for each of them. If the respondent knows more than 5 peers, then 5 peers were selected in a near-random fashion using an age-related selection process of peers with an age that was closest to the participant. The entered data (initials, last 5 digits of phone number) were not stored, and the hash ID was stored. The hashed ID could not be used to reconstruct the respondent's provided data. This information helped to evaluate the rate at which participants' networks contained other sampled participants.

During the PNS implementation, several logistical challenges emerged that could impact data completeness and accuracy. These include, some FSW participants did not own a mobile phone but possessed a SIM card to recall peer numbers needed for constructing the hash ID. To address this, the study team provided a spare phone to allow participants to access their contacts. Additionally, some FSWs did not know their peers' official names, making it difficult to accurately generate name initials for the hash ID. To mitigate this, participants were advised to attempt a mobile money transfer to the peer's number, which would reveal the name registered with the telecom provider—ensuring consistency in peer identification. These steps improved the accuracy of peer-matching and minimized duplicate entries. Other factors like stigma, mobility, and literacy were also anticipated; study staff were trained in sensitive engagement and used simplified tools to accommodate diverse literacy levels.

## Sample size and sampling

The sample size calculation was based on the FSW biobehavioral survey (BBS), which was sufficiently powered to estimate the provincial-level HIV. The sample size calculation was based on the province-specific prevalence of HIV among FSWs aged 15 years and above, estimated from the previous rounds of FSW BBS [9].

The design effect (Deff) was calculated separately for each province to account for potential clustering in sampling process, particularly due to social network-based recruitment. The estimated Deff values were as follows: 0.998 for the Eastern Province, 1.683 for the Western Province, 1.221 for the Northern Province, 3.245 for the Southern Province and 1.508 for Kigali City. These variations reflect differences in population structure and network connectivity across provinces. A Z-score of 1.96 was used for the 95% confidence level ($\alpha = 0.05$).. Additionally, the finite population correction (FPC) was applied based on the most recent FSW population size estimates from the 2018 PSE study [34], which helped improve precision in provinces with relative small estimated FSW populations. Based on these parameters, a minimum total sample size of 2,500 was determined: 415 for both Eastern and Southern Provinces, 623 for Western, 503 for Northern and 544 for City of Kigali.

## Data management

Data were collected electronically using an Open Data Kit (ODK) [39] installed on android tablets. All collected data were reviewed daily and checked for errors before submission. Daily, data from completed participant's questionnaires were electronically pushed to a password-protected database to ensure data safety.

Data quality checks were conducted regularly to ensure that high-quality data were generated. Mainly for PNS purposes, the RDS plot was run to view the recruitment graph and check if the tree matches what actually was happening in recruitment, checked for duplicated hashed IDs (duplicates are expected, but here, we checked that these are indeed unique individuals), checked for cases where a single subject reports the same hashed ID value for two of their contacts, and checked for subjects who report the exact same network contacts. To monitor and address potential seed dependence, we analyzed recruitment patterns and ensured that multiple waves of recruitment were achieved, facilitating the attainment of equilibrium and enhancing the representativeness of our sample.

To ensure high data quality and consistency, on a typical working day, data monitoring was conducted by the central study coordination team. This involved routine communication with field teams to review recruitment chain progression at each site through visualizations of recruitment trees and RDS metrics. Inconsistencies, such as duplicate hashed IDs or mismatched peer recruiter information, were flagged and addressed promptly through team debriefs and follow-up verification with participants when needed. Additionally, three complementary documentation tools were used for cross-validation: individual data collector reports, team leader summaries, and the main digital data files. The assumptions underlying the PNS methodology, such as the presence of adequate social connectivity and multiple recruitment waves, were actively monitored throughout data collection. Daily data reviews enabled the identification of recruitment stagnation or clustering, allowing timely interventions such as the activation of additional seeds.

At the end of data collection, all study site-level data in a CSV format (Comma Delimited) were merged with coupon recruitment information from RDS to track for chain referral aspects, including convergence and mitigate seed dependence, to ensure that the final sample adequately represents the broader FSW population. Anomalies were resolved by triangulating multiple study data sources. This systematic approach helped maintain data integrity and ensured that recruitment and network data reflected the intended sampling design.

## Statistical analysis

A complete dataset was composed of key PNS variables, including subject ID (unique identifiers for individuals in the dataset), recruiter ID (unique identifier for the recruiter of the subject), subject hash ID (the privatized identifier for the subject), degree (the network size of the individual, this excludes contacts for whom the individual does not know their identifiers), and contact hash IDs (privatized identifiers for each contact of the subject).

Each site-level dataset was converted into an RDS coupon data frame, which means that each recruiter is aligned alongside the corresponding direct recruits. With this data frame, each seed is regarded as the base of a tree that branches out as its recruits recruit more people, and each tree is its own sample.

Three estimators were considered, including *Cross-Sample Estimator,* which uses the overlap between each individual's friend group and the sampled individuals in other trees. The other estimator was *Cross-Alter Estimator,* which uses the overlap between each individual's friend group and friend groups from other sampled individuals who are in different trees. The third estimator was the *Cross-Network Estimator,* which combines the Cross-Sample and Cross-Alter Estimators. The intuition for PNS is "the rate at which subjects' networks contain other sampled subjects recruited by other seeds is related to population size."

The PNS method relies on three assumptions, including: connections between recruits and recruiters are completely random; small sample fractions (i.e., small sample fractions lead to potentially large amounts of sampling error when estimating population size); and long recruitment chains. However, these assumptions do not always hold, imposing a methodological limitation in that case. The *Cross-Network Estimator* displays reduced volatility compared to both the *Cross-Alter* and *Cross-Sample estimators and* showed little biased results considering different levels sample fractions and network sizes as described elsewhere [30]. Consequently, to assess estimator robustness, variance was the main parameter used.

The Cross-Network Estimator was selected as the primary estimator due to its superior performance in minimizing bias and variability. Simulation studies have shown that while the Cross-Sample estimators is prone to appreciable bias—especially at higher sampling fractions—the Cross-Network Estimator consistently demonstrates lower variance and minimal bias across diverse conditions. This is largely because it utilizes two types of matches: those between nominated alters and sampled individuals, and those between alters themselves [30]. This dual-source structure provides a stronger statistical basis for inference, particularly when studying hidden populations with complex and variable network structures, such as female sex workers (FSWs) in Rwanda. Furthermore, to produce a national population size estimate, site-level Cross-Network Estimator estimates were proportionally weighted using existing programmatic data, ensuring an aggregate figure that reflects regional distribution and service coverage.

To address potential exclusion of FSWs without active Sim card and personal cell phones, the proportion of FSWs without cell phones was estimated based on daily field reports and observations during data collection. This proportion was then used to adjust the final population size estimates, ensuring that this subgroup was appropriately represented in the results. These steps helped minimize underrepresentation of highly vulnerable FSWs who may be more socially or economically marginalized.

All data analysis processes were performed using RStudio-2023.06.2-561 with the "RDS" and "*pnspop*" packages. Confidence intervals were calculated using the bootstrap process with the number of samples set to 10,000.

Site-level estimates were aggregated at the national level, then the pooled national estimates were distributed by province using proportions of FSW from existing program data. To have provincial-level estimates, we assumed that the

provincial-level distribution of FSW population reported by HIV program in its' 2022–2023 HIV annual report remains the same for the current FSW population size estimation, therefore the pooled estimate was proportionally distributed across provinces using the percent proportion distribution from the annual HIV report 2022–2023. Regarding the PNS underlining assumptions, the sample size used was a significant portion of the previously estimated size of FSW, and a long recruitment chain was achieved by reaching 11 waves. The final analysis outputs include point estimates with corresponding 95% credible intervals. Finally, each estimate was adjusted for the proportion of FSW who did not have cell phones.

### Ethical consideration

The survey was reviewed and approved by the Rwanda National Ethics Committee (RNEC), protocol ID: IRB 00001497 of IORG0001100. This study ensured maximum anonymity and confidentiality to guarantee that study participants are not victimized for participation. In this line, no names or identifying information was collected from any survey participant, participants were only identified using unique IDs. Furthermore, the study sites were secure places within a health facility setting that usually offers HIV services to FSW community to minimize possibility of stigma. Finally, paper-based study documents were maintained by the team leaders and stored in a designated locked cabinet during field work. Access to data was restricted and closely monitored, and all electronic data collection tools were password protected.

Participation in the study was voluntarily and free to withdraw at any time during the conduct of the study. Neither refusal to participate nor withdrawal will affect services they would normally receive. In Rwanda, children under 18 years require parental consent prior to participation in the survey. The only exception is for "emancipated minors" who are children head of household; these children are not required to provide the consent of a caregiver but can instead consent directly. For FSWs under the age of consent who are NOT the head of household, we request a waiver of informed consent and incorporated safeguards, including referrals to child protection services, in line with international ethical guidelines for research involving minors [41].

A written informed consent was obtained from the study participants to be part of the survey. Furthermore, a waiver of informed consent for participants aged 15–17- years was granted by RNEC. Children <18 years of age identified as being engaged in sex work, trafficked, or victim of violence, and were referred for appropriate services to ensure their protection and well-being.

Data collection staff completed training on human subjects' research and signed a confidentiality agreement before the start of enrollment. Participants were compensated with 5,000 Rwf (4.24 USD) for transportation costs and time during their first visit and 2,000 Rwf (1.70 USD) for the second visit as compensation for time and 1,000 Rwf (0.85 USD) for each successful referral enrolled in the survey. Compensation for transport was determined based on the areas' transport cost as stipulated by Rwanda Utilities Regulatory Authority (RURA) and delivered in cash by the study site accountant.

Prior to implementation, field staff received a one-week standardized training together in one site, followed by a half-day refresher training at their respective sites. These trainings focused on general knowledge of FSW, ethical issues in human subject research, and standard operating procedures for the RDS survey and PNS implementation.

### Results

In total, 30 FSWs were enrolled as seeds across 10 study sites countrywide to initiate referral chain recruitment. The maximum wavelength achieved during recruitment was 11 with a mode of 4. Fig 1 below illustrates the RDS recruitment tree by province.

The highest proportion of FSW were estimated among those aged 30–39 years, with the smallest proportion among 15–17 age group (0.8%) which was anticipated. The results shows that the study recruitment tapped into all age categories of FSW, which is a great indication for a representative sample as far as age is concerned. Regarding marital status, the majority, 69.8% (95% CI: 66.5-72.9), 22.0% (95% CI:19.3-24.9) were single and divorced/separated respectively, and the same distribution remains across all provinces. In terms of where FSW meet or find

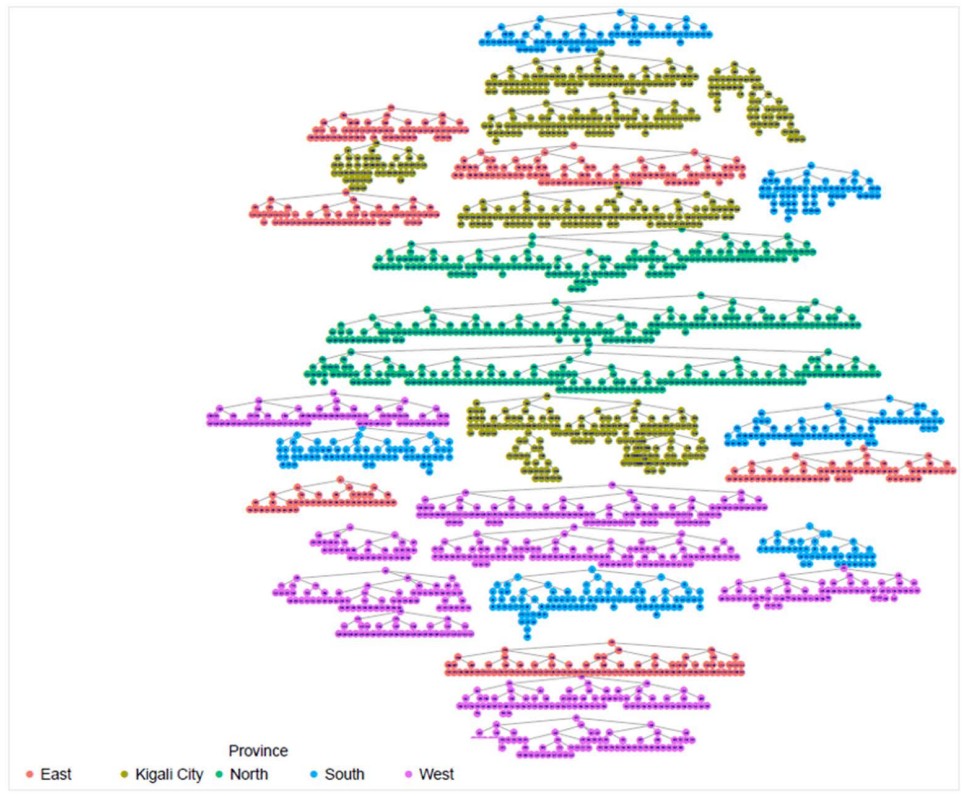

**Fig 1. Recruitment tree by province, Rwanda FSW PNS 2023.**

clients, FSW participants presented diverse ways including previously untapped subgroups when using venue-based sampling approaches as Time Location Sampling (TLS). in this line, an estimated 14.3% (95% CI: 12.3-16.6) of FSW meet clients using the internet, phone brokers, or escort agency. Of all recruited FSW, 79.4% (95% CI: 76.4 −82.2) had a cell phone and a Sim card, and 81.4% (95% CI: 78.0–84.4) of them had at least nominated an FSW peer of whom she knows by name and has a telephone contact. This aligns with what formative assessment (FA) finds and provides confidence in the suggested way of creating the anonymized unique identification using the last 5 digits of one's phone number and name's initials combination to trace deidentified recruitment chains (see Table 1 and Figs 2–6).

Site-level population size estimate variation was observed across estimators. The population size estimates produced by the Cross-Sample estimator were greater than those produced using the Cross-Alter estimator at the Gihundwe HC, Kibuye HC, Muhoza HC, and Mukarange HC study sites. Population size estimates from Gisenyi HC and Gitarama HC remained almost the same across the two estimators. However, population size estimates produced by the Cross-Sample estimator were less than those produced using the Cross-Alter estimator for the Nyagatare HC, City of Kigali, and Rango HC study sites. Site-level population size estimates produced using the Cross-Network estimator were observed to be between the estimates produced using Cross-Sample and Cross-Alter estimators except for the Gisenyi HC study site. The same patterns were observed for both adjusted and unadjusted population size estimates; however, the differences are not statistically significant (see Table 2). Site-level estimates present high volatility; thus, their use and interpretation should be done cautiously.

**Table 1. Social demographic characteristics of participants by province, Rwanda FSW PNS 2023.**

| | N | Province | | | | | |
|---|---|---|---|---|---|---|---|
| | | North | East | West | City of Kigali | South | Overall |
| | | Row % [95% CI] | Row % [95% CI] | Row % [95% CI] | Row % [95% CI] | Row % [95% CI] | Row % [95% CI] |
| **Age group** | | | | | | | |
| 15–17 | 23 | 2.6 [1.4–4.8] | 0.3 [0.1–1.5] | 1.5 [0.5–4.1] | 0.4 [0.1–1.7] | 0.0 [0.0–0.2] | 0.8 [0.5–1.3] |
| 18–24 | 515 | 26.3 [22.1–30.9] | 21.3 [16.5–27.1] | 25.4 [20.8–30.7] | 24.5 [18.3–31.9] | 13.6 [9.6–19.1] | 20.7 [17.9–23.9] |
| 25–29 | 497 | 23.5 [19.4–28.0] | 22.4 [17.3–28.5] | 20.8 [16.8–25.5] | 12.9 [9.1–18.0] | 19 [14.3–24.7] | 18.1 [15.7–20.8] |
| 30–34 | 535 | 22.2 [18.1–26.9] | 21.7 [16.6–27.9] | 19.8 [15.9–24.3] | 15.8 [12.3–20.2] | 26.3 [20.2–33.5] | 21.4 [18.6–24.6] |
| 35–39 | 502 | 15.0 [11.9–18.9] | 17.7 [13.4–22.9] | 19.6 [15.6–24.3] | 20.0 [15.6–25.1] | 23.9 [18.3–30.6] | 20.4 [17.7–23.4] |
| 40+ | 439 | 10.5 [7.8–13.9] | 16.6 [12.3–22.0] | 12.9 [9.9–16.8] | 26.4 [20.9–32.8] | 17.1 [12.6–22.8] | 18.6 [15.9–21.5] |
| **Current marital status** | | | | | | | |
| Single | 1733 | 73.3 [68.5–77.5] | 74.9 [69.0–80.0] | 77.8 [73.0–81.9] | 55.0 [48.3–61.5] | 78.7 [72.4–83.9] | 69.8 [66.5–72.9] |
| Married/Cohabitating | 54 | 0.3 [0.0–2.4] | 0.8 [0.3–2.5] | 1.3 [0.3–5.0] | 9.3 [6.6–13.0] | 0.7 [0.2–2.6] | 3.5 [2.5–4.8] |
| Divorced/Separated | 633 | 24.5 [20.4–29.1] | 22.0 [17.1–27.7] | 16.4 [13.0–20.5] | 27.8 [22.6–33.6] | 16.8 [12.2–22.8] | 22.0 [19.3–24.9] |
| Widow | 88 | 1.9 [0.9–4.0] | 2.3 [1.1–4.8] | 4.2 [2.4–7.5] | 7.9 [4.5–13.3] | 3.5 [1.7–7.0] | 4.6 [3.2–6.7] |
| Prefer not to answer | 3 | 0 | 0 | 0.3 [0.1–1.4] | 0 | 0.3 [0.0–2.1] | 0.1 [0.0–0.7] |
| **Education level** | | | | | | | |
| None | 490 | 16.7 [13.2–20.9] | 25.9 [20.4–32.4] | 22.9 [18.4–28.2] | 16.5 [12.5–21.4] | 18 [13.1–24.2] | 18.0 [15.5–20.8] |
| Primary | 1327 | 43.5 [38.5–48.5] | 50.7 [44.1–57.2] | 49.1 [43.7–54.5] | 55.1 [48.4–61.7] | 59.8 [52.6–66.6] | 54.0 [50.5–57.6] |
| Secondary/vocational/higher education | 662 | 39.8 [34.9–44.9] | 23.4 [18.4–29.3] | 24.5 [20.2–29.3] | 28.4 [22.7–34.9] | 20.5 [15.2–27.0] | 27.1 [24.0–30.3] |
| Do not know/No answer | 32 | 0 | 0 | 3.5 [1.9–6.4] | 0 | 1.7 [0.6–4.8] | 0.9 [0.4–1.9] |
| **Where do you usually meet or find clients?** | | | | | | | |
| Brothel/Guesthouse/Massage/Parlor | 118 | 0.2 [0.0–1.1] | 1.8 [0.6–5.1] | 4.8 [3.2–7.0] | 16.2 [12.2–21.3] | 3.2 [1.5–6.5] | 6.9 [5.4–8.9] |
| Hotel/Club/Bar/Restaurant | 1110 | 47.8 [42.8–52.9] | 54.6 [48.1–61.1] | 58 [52.6–63.2] | 36.1 [29.4–43.3] | 75.5 [69.5–80.6] | 55.3 [51.8–58.8] |
| Street/Park | 720 | 22.2 [18.3–26.7] | 7.5 [5.6–9.9] | 21.2 [17.1–25.9] | 34.5 [28.8–40.8] | 8.3 [5.3–12.8] | 20.3 [17.8–23.1] |
| Other public places | 66 | 2.8 [1.3–6.2] | 4.4 [2.1–9.0] | 1.4 [0.6–3.0] | 1.4 [0.6–3.2] | 2.0 [1.0–4.0] | 2.0 [1.4–3.0] |
| Internet, phone broker, escort agency | 467 | 26 [21.7–30.8] | 30.1 [24.4–36.6] | 12.4 [9.3–16.3] | 11.5 [8.0–16.3] | 9.6 [6.6–13.7] | 14.3 [12.3–16.6] |
| Other | 29 | 1.0 [0.2–4.3] | 1.6 [0.6–4.3] | 2.2 [1.0–4.7] | 0.2 [0.1–0.4] | 1.5 [0.6–3.8] | 1.0 [0.6–1.9] |
| Prefer not to answer | 1 | 0 | 0 | 0.1 [0.0–0.9] | 0 | 0 | 0.0 [0.0–0.1] |
| **Have a Sim card and cell phone** | | | | | | | |
| Yes | 2002 | 75.2 [70.5–79.3] | 95.9 [94.3–97.1] | 76.1 [70.8–80.7] | 86.6 [82.1–90.1] | 74.0 [67.2–79.9] | 79.4 [76.4–82.2] |
| No | 509 | 24.8 [20.7–29.5] | 4.1 [2.9–5.7] | 23.9 [19.3–29.2] | 13.4 [9.9–17.9] | 26.0 [20.1–32.8] | 20.6 [17.8–23.6] |
| **Has at least one peer's contact** | | | | | | | |
| Yes | 1509 | 93.6 [88.3–96.6] | 91.5 [89.2–93.4] | 59.6 [53.5–65.5] | 64.3 [56.6–71.4] | 96.0 [94.5–97.1] | 81.4 [78.0–84.4] |
| No | 493 | 6.4 [3.4–11.7] | 8.5 [6.6–10.8] | 40.4 [34.5–46.5] | 35.7 [28.6–43.4] | 4.0 [2.9–5.5] | 18.6 [15.6–22.0] |
| **Number of peers from whom contacts are available** | | | | | | | |
| One peer | 188 | 3.7 [1.9–7.2] | 2.4 [1.4–3.9] | 15.7 [10.4–23.0] | 37.3 [28.7–46.7] | 1.8 [1.1–2.9] | 12.9 [9.9–16.6] |
| Two peers | 270 | 6.8 [4.5–10.1] | 1.7 [0.9–3.3] | 26.5 [20.9–33.0] | 32.8 [25.6–41.0] | 2.7 [1.8–4.1] | 13.2 [10.9–16.0] |
| Three peers | 548 | 36.6 [31.1–42.4] | 95.6 [93.5–97.0] | 37.4 [30.3–45.2] | 16.6 [11.7–23.1] | 4.1 [2.5–6.9] | 21.3 [18.8–24.2] |
| Four Peers | 128 | 16 [12.0–21.1] | 0.2 [0.0–1.2] | 9.9 [6.6–14.6] | 7.1 [4.3–11.5] | 5.8 [2.9–11.4] | 8.1 [6.2–10.5] |
| Five peers | 238 | 22.1 [17.6–27.4] | 0.1 [0.0–0.3] | 8.3 [5.3–12.9] | 3.3 [1.6–6.6] | 67.2 [59.3–74.3] | 33.1 [28.7–37.8] |
| More than five peers | 137 | 14.8 [11.3–19.1] | 0.0 [0.0–0.1] | 2.2 [1.0–4.6] | 2.9 [1.3–6.3] | 18.3 [12.7–25.7] | 11.3 [8.7–14.6] |

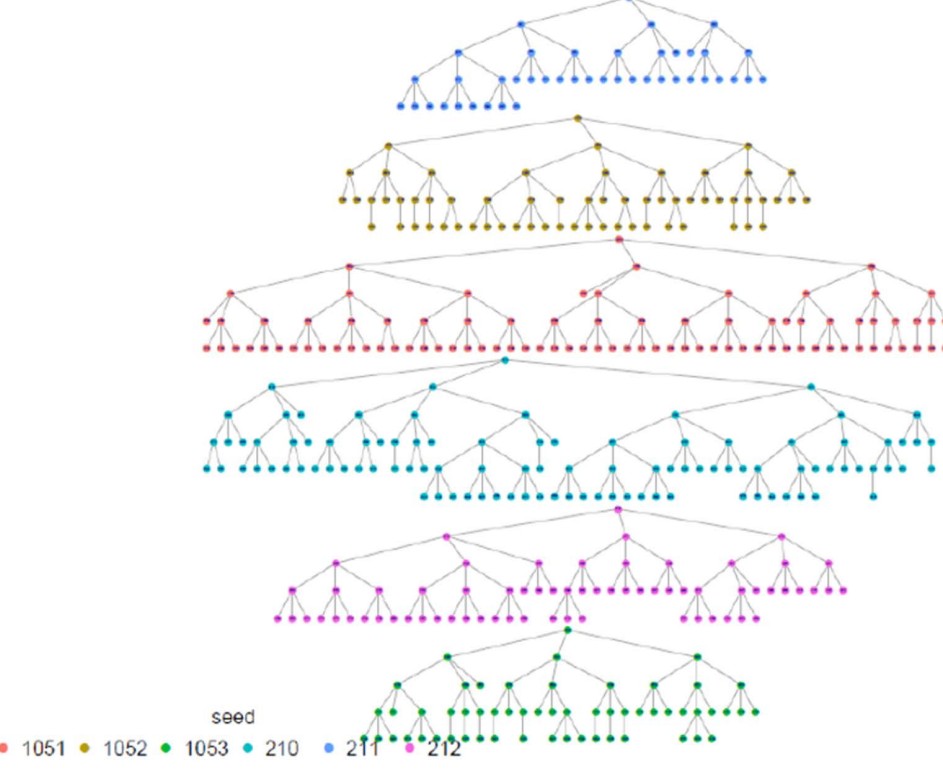

**Fig 2. Eastern Province recruitment tree, Rwanda FSW PNS 2023.**

Table 3 presents aggregated site-level population size estimates as a pooled estimate by the estimator used to produce the estimate.

Looking at the pooled population size estimate, the Cross-Sample estimator tends to produce larger estimates as compared to the Cross-Alter and Cross-Network estimators considering both unadjusted and adjusted estimates. On the other hand, population size estimates produced using Cross-Alter and Cross-network estimators go hand in hand considering both adjusted and unadjusted estimates (see Table 3). The observed patterns in estimates across estimators aligns with several simulation study finding where Cross-Alter and Cross-Network estimators present similar level of performance, but Cross-Alter shows an elevated variance level [30].

Table 4 presents provincial-level adjusted FSW population size estimates produced using the Cross-Network estimator, which combines both Cross-Sample and Cross-Alter estimator features and presents the lowest variance and bias as compared to other estimators.

The national level FSW PSE was estimated at 98,587 (95% CI: 82,978–114,196), corresponding to 2.3% of the total adult female population aged 15 years and above in Rwanda. The highest FSW PSE was observed in the City of Kigali with 31,018 (95% CI: 26,107–35,929), followed by West province with 20,593 (95% CI: 17,332–23,853), the East province with 19,833 (95% CI: 16,693–22,973), and the South province with 15,826 (95% CI: 13,320–18,331). The lowest FSW population size was estimated in the North province with 11,317 (95% CI: 9,526–13,109). These patterns align with busy economic and social locations in Rwanda, where most activities that attract female sex worker are concentrated, including but not limited to areas associated with commerce, tourism and leisure.

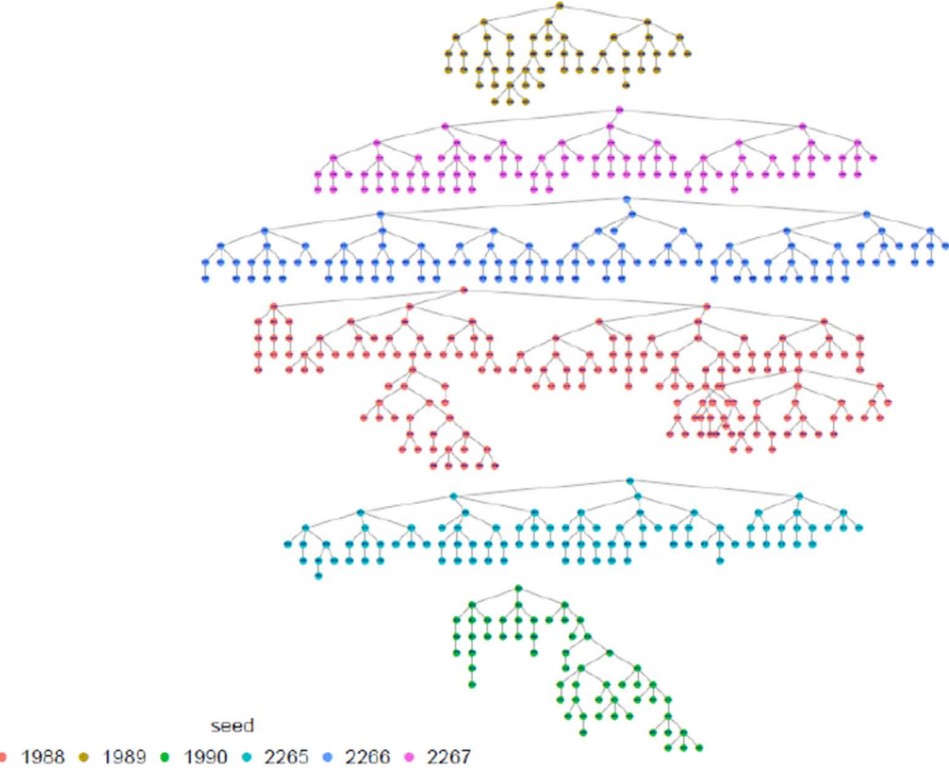

**Fig 3. City of Kigali recruitment tree, Rwanda FSW PNS 2023.**

## Discussion

The national population size of female sex workers and sexually exploited minors aged 15 and above in Rwanda was estimated at 98,587 (95% CI: 82,978–114,196), corresponding to 2.3% of the total females aged 15 years and above in Rwanda based on the 5th Rwanda Housing and Population census, 2022 [42]. The highest FSW concentration was found in the City of Kigali, with 5.3% [4.5–6.1] and the lowest in the North province with 1.6% [1.3–1.9] as % of female aged 15 years and above of the general population.

Due to persistent stigma around sex work, FSWs often operate with a high level of secrecy and may use emerging technologies such as encrypted messaging apps or private social media platforms. for privacy. Rwanda has been experiencing a generalized HIV prevalence over the last decade, with a concentrated HIV epidemic among FSWs[9], where HIV prevalence is more than tenfold that of the general population [43], making them to be considered among key populations for HIV programs. Having an estimate of the population size of FSW would inform programs and interventions to better plan, monitor, and prevent disease spillovers into the general population.

To date, there have been four rounds of studies aimed at estimating the population size of FSW in Rwanda, with the commonality of all using Time Location Sampling (TLS) methodologies [33,34,44]. Capture-recapture, enumeration, and multiplier were listed among the methods used, with a commonly stated methodological limitation of inability to tackle within non-venue-based FSW and leading to potential underestimation of FSW. In contrast, in this survey we used PNS, a PSE that is capable of reaching different FSW subgroups, including non-venue-based subgroups

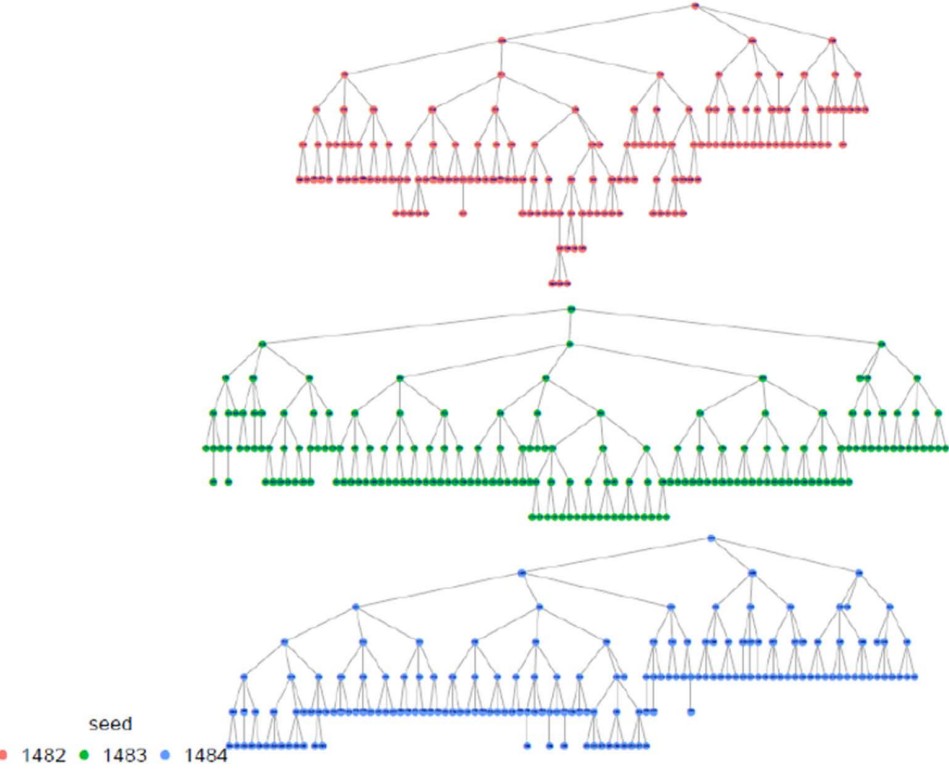

seed
● 1482 ● 1483 ● 1484

**Fig 4. Northern province recruitment tree, Rwanda FSW PNS 2023.**

[30], and produces financial resources effective and more credible inferences about population size, given that it is built into RDS.

The reported FSW population size estimate looks larger when compared with previously estimated size of FSW population in Rwanda. The variances between the current FSW PSE and the previous ones can be more potentially attributed to the methodological capability to reach non-venue based FSWs around the country. The National HIV annual report 2022−23 reported a total number of 60,460 FSW identified during the reporting year period in the HIV program [45]. Believing that the program reported number is not exhaustive and recognizing the limitations associated with program data, including the possible inability to deidentify individual-level data, give more confidence in the PSE resulted from the current study.

This study successfully reached various subgroups of FSWs, including non-venue-based individuals, with 14.3% of respondents reporting solicitation through internet platforms, phone brokers, or escort agencies, as shown in Table 1. This reflects the evolving nature of sex work beyond physical venues and highlights the adaptability of the PNS methodology to reach these populations.

Important limitations associated to the use of PNS to estimate the population size in our context, is related to the coverage of SIM cards and cell phones among FSW within the survey sample, which was used to produce hashed ids to uniquely identify overlaps (alters) in the tracing network sampling. Access to mobile technology varies among individuals, and not all FSWs may possess a SIM card and a functioning cell phone that might lead to a potential underrepresentation. To mitigate this, we quantified the proportion of participants without phones and applied this proportion as a correction

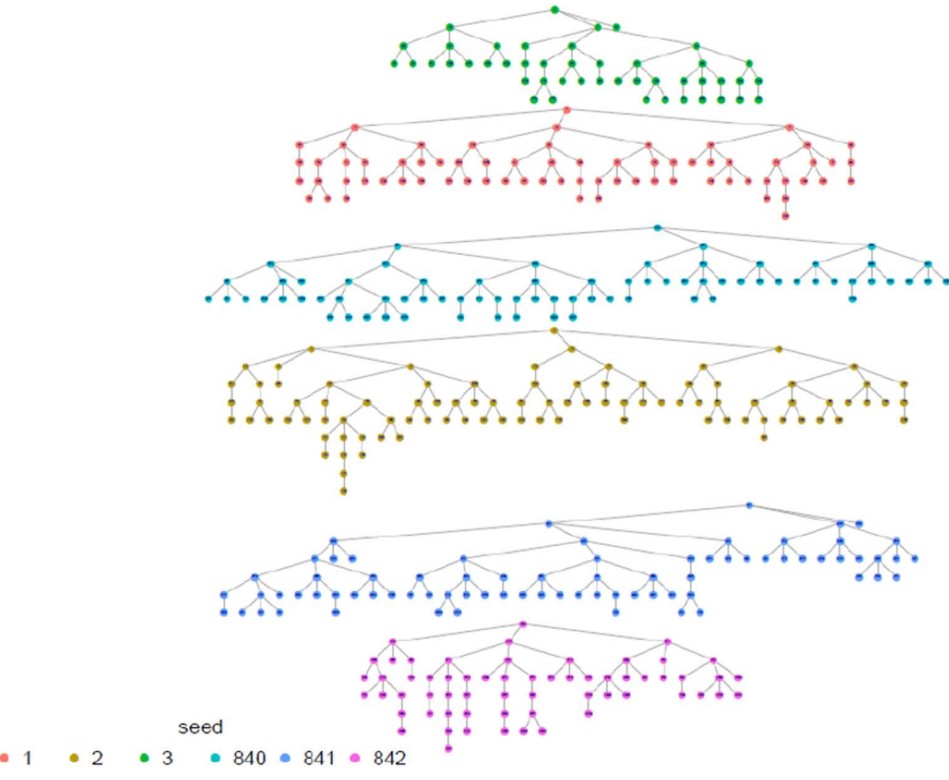

seed
● 1  ● 2  ● 3  ● 840  ● 841  ● 842

**Fig 5. Southern province recruitment tree, Rwanda FSW PNS 2023.**

factor to adjust the final population size estimates. Although this adjustment helped reduce bias, its exact influence on the credibility intervals remains difficult to quantify.

Furthermore, the estimates from this study might have been affected by those FSW who decide not to provide FSW phone numbers, the data collection tool used that did enable to delimit participants only for the province where the survey was being implemented, and the social desirability bias of the BBS. also, we acknowledge methodological related limitations in line with underlining PNS assumptions. Lastly, we recognize that our estimates may underrepresent the full population of sex workers by excluding individuals who engage in sex work part-time or intermittently and do not consider it their main source of income.

With these estimates of the population size of the female sex worker, policymakers will be able to set realistic targets and goals aimed at reducing HIV transmission and improving health outcomes among this group. Furthermore, these estimates provide a baseline against which progress will be measured, enabling the monitoring and evaluation of interventions aimed at reaching and serving female sex workers effectively. And in addition, the data from this study will inform resource mobilization and allocation efforts, ensuring that sufficient funding and support are directed towards HIV prevention and treatment programs tailored to the needs of female sex workers.

In conclusion, this study sheds light on critical aspects of the female sex workers (FSW) population in Rwanda, revealing a higher concentration compared to the regional average, with 2.3% identified as FSWs of the total adult females in the general population in contrast to the 1.1% reported in sub-Saharan Africa [46]. FSW population size estimate derived from this study serves as the basis for targeted interventions, resource allocation, advocacy efforts, resources

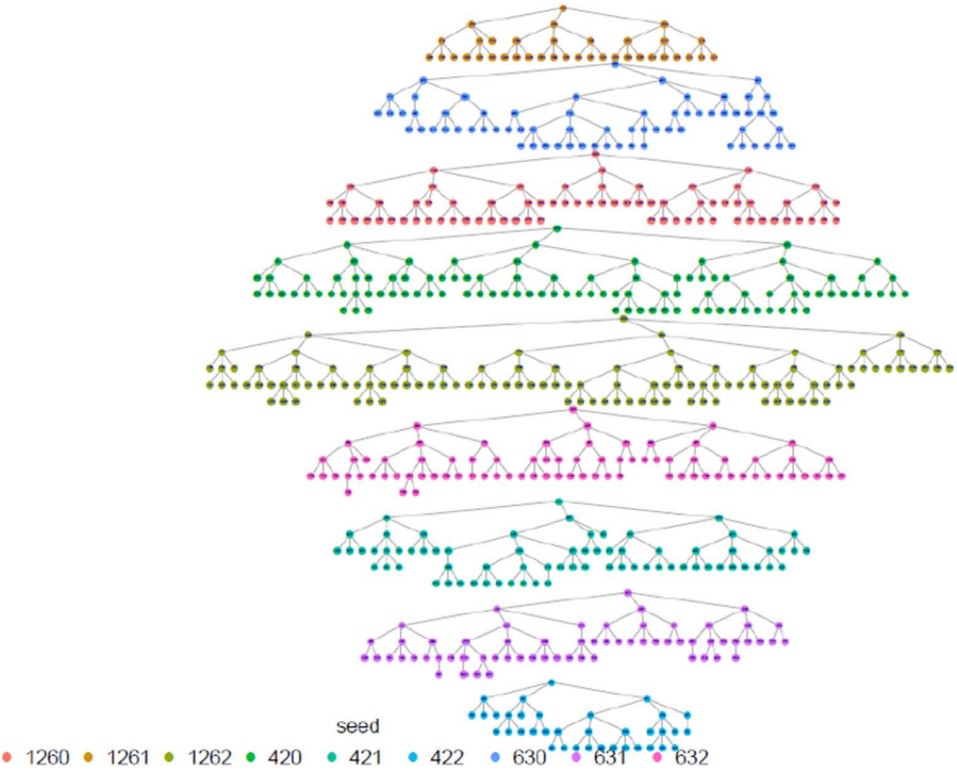

**Fig 6. Western province recruitment tree, Rwanda FSW PNS 2023.**

mobilization, and policy development aimed at improving access to preventive, care and treatment services for FSW in Rwanda. Additionally, the findings from this study provide critical insights for improving geographic targeting and resource allocation for HIV prevention and care among FSWs in Rwanda. In addition to informing differentiated service delivery. Furthermore, this is the first time that PNS is implemented for the estimation of the FSW population size estimate in Rwanda, adding to the emerging tools that we have in the hard-to-reach PSE field. Moving forward, future research endeavors could explore smaller area estimation techniques to ascertain the distribution of FSWs at the district level, enabling more localized and tailored interventions.

The findings from this study underscore the need for future research to address methodological and contextual limitations encountered. One key limitation was the dependency of PNS on mobile technology, specifically, SIM card and cell phone access for constructing hashed IDs to identify unique alters. While we applied a correction factor based on the quantified proportion of participants without phones, the precise impact on credibility intervals remains difficult to estimate. Future studies should explore alternative or supplementary approaches to hashed ID generation. Methodological enhancements should also include critical validation of PNS assumptions, such as randomness in peer nomination and adequate network penetration to better understand their applicability in other similar contexts. Finally, future efforts should consider longitudinal and mixed-methods studies to capture the dynamics of part-time or hidden sex workers who may not identify as FSWs or engage in sex work intermittently, thereby enhancing both population coverage and the validity of resulting estimates.

**Table 2. Unadjusted and adjusted study site-level population size estimates of female sex workers by estimation method.**

| Study site | Estimators | Unadjusted Estimates | | | Adjusted Estimates | | |
|---|---|---|---|---|---|---|---|
| | | PSE | Lower Bound | Upper Bound | PSE | Lower Bound | Upper Bound |
| GIHUNDWE HC | Cross-Sample | 15,538 | 3,399 | 71,023 | 19,599 | 4,288 | 89,586 |
| GISENYI HC | Cross-Sample | 1,463 | 1,033 | 2,073 | 1,494 | 1,054 | 2,116 |
| GITARAMA HC | Cross-Sample | 6,256 | 2,884 | 13,571 | 7,839 | 3,613 | 17,005 |
| KIBUYE HC | Cross-Sample | 5,875 | 2,995 | 11,527 | 8,728 | 4,448 | 17,123 |
| MUHOZA HC | Cross-Sample | 104,226 | 41,554 | 261,421 | 140,369 | 55,964 | 352,075 |
| MUKARANGE HC | Cross-Sample | 1,021 | 773 | 1,349 | 1,371 | 1,038 | 1,811 |
| NYAGATARE HC | Cross-Sample | 3,905 | 1,945 | 7,842 | 3,905 | 1,945 | 7,842 |
| RANGO HC | Cross-Sample | 2,903 | 2,151 | 3,919 | 3,852 | 2,854 | 5,200 |
| CITY OF KIGALI | Cross-Sample | 11,314 | 8,029 | 15,942 | 14,414 | 10,229 | 20,310 |
| GIHUNDWE HC | Cross-Alter | 4,043 | 2,975 | 5,494 | 5,099 | 3,752 | 6,930 |
| GISENYI HC | Cross-Alter | 1,511 | 1,159 | 1,969 | 1,542 | 1,183 | 2,009 |
| GITARAMA HC | Cross-Alter | 6,006 | 3,830 | 9,418 | 7,526 | 4,799 | 11,801 |
| KIBUYE HC | Cross-Alter | 3,696 | 2,830 | 4,827 | 5,490 | 4,203 | 7,170 |
| MUHOZA HC | Cross-Alter | 27,472 | 21,272 | 35,479 | 36,999 | 28,649 | 47,783 |
| MUKARANGE HC | Cross-Alter | 354 | 226 | 556 | 476 | 303 | 746 |
| NYAGATARE HC | Cross-Alter | 10,166 | 5,196 | 19,892 | 10,166 | 5,196 | 19,892 |
| RANGO HC | Cross-Alter | 4,650 | 3,580 | 6,039 | 6,170 | 4,750 | 8,013 |
| CITY OF KIGALI | Cross-Alter | 18,457 | 13,978 | 24,373 | 23,514 | 17,807 | 31,051 |
| GIHUNDWE HC | Cross-Network | 5,479 | 3,996 | 7,512 | 6,911 | 5,040 | 9,476 |
| GISENYI HC | Cross-Network | 1,491 | 1,193 | 1,863 | 1,522 | 1,218 | 1,901 |
| GITARAMA HC | Cross-Network | 6,106 | 4,117 | 9,055 | 7,651 | 5,159 | 11,346 |
| KIBUYE HC | Cross-Network | 4,241 | 3,343 | 5,381 | 6,300 | 4,965 | 7,993 |
| MUHOZA HC | Cross-Network | 32,758 | 25,288 | 42,434 | 44,117 | 34,057 | 57,149 |
| MUKARANGE HC | Cross-Network | 597 | 454 | 785 | 801 | 610 | 1,053 |
| NYAGATARE HC | Cross-Network | 7,200 | 4,364 | 11,881 | 7,200 | 4,364 | 11,881 |
| RANGO HC | Cross-Network | 4,191 | 3,351 | 5,243 | 5,562 | 4,446 | 6,957 |
| CITY OF KIGALI | Cross-Network | 14,540 | 11,991 | 17,630 | 18,523 | 15,277 | 22,460 |

**Table 3. Overall population size estimates by estimator, Rwanda FSW BBS 2023.**

| Estimators | Unadjusted Estimates | | | Adjusted Estimates | | |
|---|---|---|---|---|---|---|
| | PSE | Lower Bound | Upper Bound | PSE | Lower Bound | Upper Bound |
| Cross-Sample | 152,502 | 64,763 | 388,667 | 201,570 | 85,434 | 513,067 |
| Cross-Alter | 76,354 | 55,045 | 108,046 | 96,981 | 70,643 | 135,394 |
| Cross-Network | 76,603 | 58,097 | 101,782 | 98,587 | 82,978 | 114,196 |

## Acknowledgments

We express our gratitude to all the study participants and the team of co-investigators who helped make this work possible. Additionally, we would like to convey our sincere gratitude to the CDC team for their crucial technical support during the entire study's implementation.

**Table 4. Cross-network estimator's population size estimates by province, Rwanda FSW BBS 2023.**

| Province | Consensus Population Size Estimate [95%CI] | *PSE as % of female aged 15+years of the general population |
|---|---|---|
| North | 11,317 [9,526 −13,109] | 1.6% [1.3−1.9] |
| South | 15,826 [13,320 −18,331] | 1.7% [1,4−1.9 |
| East | 19,833 [16,693 −22,973] | 1.7% [1.4−2.0] |
| West | 20,593 [17,332 −23,853] | 2.2% [1.8−2.5] |
| City of Kigali | 31,018 [26,107−35,929] | 5.3% [4.5−6.1] |
| **Total** | **98,587 [82,978−114,196]** | **2.3% [1.9−2.6]** |

*Calculated based on the 5th Rwanda Housing and Population census, 2022 [42].

## Author contributions

**Conceptualization:** Elysee TUYISHIME, Eric REMERA.

**Data curation:** Elysee TUYISHIME.

**Formal analysis:** Elysee TUYISHIME.

**Funding acquisition:** Catherine Kayitesi, Eric REMERA.

**Investigation:** Catherine Kayitesi, Eric REMERA.

**Methodology:** Elysee TUYISHIME.

**Project administration:** Catherine Kayitesi.

**Resources:** Catherine Kayitesi, Eric REMERA.

**Supervision:** Eric REMERA, Ignace Habimana Kabano, Angela Unna Chukwu.

**Validation:** Catherine Kayitesi, Eric REMERA, Samuel Sewava Malamba, Angela Unna Chukwu.

**Visualization:** Elysee TUYISHIME.

**Writing – original draft:** Elysee TUYISHIME.

**Writing – review & editing:** Catherine Kayitesi, Eric REMERA, Samuel Sewava Malamba, Ignace Habimana Kabano, Angela Unna Chukwu.

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
