## [Decision Letter · Decision Letter 0]

16 Feb 2024

PONE-D-23-42748Population size estimation of female sex workers and sexually exploited minors using privatized network sampling, Rwanda, 2023.PLOS ONE

Dear Dr. TUYISHIME,

Thank you for submitting your manuscript to PLOS ONE. After careful consideration, we feel that it has merit but does not fully meet PLOS ONE’s publication criteria as it currently stands. Therefore, we invite you to submit a revised version of the manuscript that addresses the points raised during the review process.

**ACADEMIC EDITOR: **

This paper is publishable upon addressing the comments provided by our peer reviewers as outlined below.

We look forward to receiving your revised manuscript.

Kind regards,

Maureen Musimbi Akolo, Ph.D

Academic Editor

PLOS ONE

3. For studies involving third-party data, we encourage authors to share any data specific to their analyses that they can legally distribute. PLOS recognizes, however, that authors may be using third-party data they do not have the rights to share. When third-party data cannot be publicly shared, authors must provide all information necessary for interested researchers to apply to gain access to the data. (https://journals.plos.org/plosone/s/data-availability#loc-acceptable-data-access-restrictions)

a) A description of the data set and the third-party source

b) If applicable, verification of permission to use the data set

c) Confirmation of whether the authors received any special privileges in accessing the data that other researchers would not have

d) All necessary contact information others would need to apply to gain access to the data

Reviewers' comments:

Reviewer's Responses to Questions

**Comments to the Author**

1. Is the manuscript technically sound, and do the data support the conclusions?

Reviewer #1: Yes

Reviewer #2: Yes

2. Has the statistical analysis been performed appropriately and rigorously? 

Reviewer #1: Yes

Reviewer #2: Yes

3. Have the authors made all data underlying the findings in their manuscript fully available?

Reviewer #1: Yes

Reviewer #2: No

4. Is the manuscript presented in an intelligible fashion and written in standard English?

Reviewer #1: Yes

Reviewer #2: Yes

5. Review Comments to the Author

Reviewer #1: Introduction

The research article titled “Population size estimation of female sex workers and sexually exploited minors using privatized network sampling, Rwanda, 2023” aims to estimate the population size of female sex workers (FSW) and sexually exploited minors in Rwanda using privatized network sampling (PNS), a method that uses network information collected within a bio-behavioral survey that used respondent-driven sampling (RDS). The study collected data from 2,511 FSW across 10 study sites in Rwanda from May 8th to June 24th, 2023. Three PNS estimators were used to calculate the population size: Cross-Sample, Cross-Alter, and Cross-Network. The national-level FSW population size was estimated at 98,587 (95% CI: 82,978 – 114,196), corresponding to 2.3% of the total adult female population aged 15 years and above in Rwanda. The highest proportion of FSW was found in the City of Kigali (5.3%), followed by the West Province (2.2%), and the lowest in the North Province (1.6%). The study concludes that PNS provides fundamental information for designing, planning, and implementing programs for FSW at the provincial level in Rwanda.

Comments for improvement

• Title: the title could benefit from greater clarity, ethical consideration, and sensitivity to the populations studied. A suggested title "Estimating the Size of Vulnerable Populations: A Comprehensive Study on Female Sex Workers and Sexually Exploited Minors in Rwanda using Private Network Sampling in 2023".

• Abstract: The abstract provides a good overview of the background, methods, results, and conclusion of the study. However, it could be improved by:

o Providing some context and motivation for why estimating the size of FSW population is important for HIV prevention and control in Rwanda.

o Stating the main research question or objective of the study.

o Avoiding repetition of information, such as mentioning the study period twice or the provincial distribution of FSW twice.

• Introduction: The introduction provides a good background and literature review on the HIV situation among FSW in Rwanda and globally, and the challenges and methods of estimating their population size. However, it could be improved by:

o Providing a clear statement of the research question or objective at the end of the introduction, and how it relates to the existing knowledge gap or research need.

o Providing some hypotheses or expectations about the results of the study, based on the literature review or the formative assessment.

• Methods: The methods section describes the study design, procedures, sample size, data management, and statistical analysis in detail and with clarity. However, it could be improved by:

o Providing more information about the ethical considerations, such as how the informed consent was obtained, how the confidentiality and anonymity of the participants were ensured, and how the compensation was determined and delivered.

o Providing more information about the formative assessment, such as the number and characteristics of the FSW who participated in the group meeting and the FGD, the main findings or themes that emerged from the FA, and how they informed the design and implementation of the PNS.

o Providing more information about the statistical analysis, such as the assumptions and limitations of the PNS estimators, and the sensitivity analysis or robustness checks performed

• Results:

o Provide more interpretation and explanation of the results, such as the possible reasons or factors for the observed patterns, trends, or differences in the data, and the implications or significance of the results for the research question or objective.

• Discussion: The discussion section summarizes and discusses the main findings and implications of the study and compares and contrasts them with the existing literature. However, it could be improved by:

o Providing more contextual and practical implications of the results, such as the policy and program recommendations for improving the access and quality of HIV prevention and treatment services for FSW in Rwanda, and the potential impact or benefit of the study for the HIV epidemic control and public health outcomes in the country.

o Providing more suggestions for future research, such as the limitations or gaps that need to be addressed, the questions or hypotheses that need to be tested, the methods or data that need to be improved or developed, the collaborations or partnerships that need to be established or strengthened, and the assumptions or uncertainties of the PNS method, the representativeness or quality of the data, or the generalizability or applicability of the findings.

• Conclusion: The conclusion section provides a brief and concise summary of the main findings, implications, and recommendations of the study. However, it could be improved by:

o Emphasizing the key messages or takeaways of the study and provide some directions or priorities for future research, such as the areas that need further investigation, the methods or data that need refinement or innovation, or the stakeholders or partners that need engagement or support.

• References:

o Check the accuracy and completeness of the citation information, such as the authors, titles, journals, volumes, issues, pages, years, DOIs.

o Check the consistency and correctness of the citation format, such as the punctuation, capitalization, italicization, or abbreviation of the elements of the citations.

Reviewer #2: Abstract's conclusion: it is difficult to locate the exact finding in the conclusion; authors are advised to state the exact finding.

Method: Ethical clearance ID (i.e. IRB ID) is needed.

Results: Just wondering whether an actual sample size of 30 is feasible enough to generalize findings to the general population.

Even though bootstrapping created many simulated samples to work with, it was not stated or acknowledged...making it difficult to relate a sample size of just 30 to 'many' thousands of samples. Kindly rectify.

Aside confidence interval generation, were authors also interested in reducing bias and performing hypothesis testing as well?

6. PLOS authors have the option to publish the peer review history of their article (what does this mean? ). If published, this will include your full peer review and any attached files.

**Do you want your identity to be public for this peer review?** For information about this choice, including consent withdrawal, please see our Privacy Policy .

Reviewer #1: **Yes: ** ABAH ISAAC OKOH

Reviewer #2: No

---

## [Author Response · Author response to Decision Letter 1]

26 Feb 2024

Responses to ACADEMIC EDITOR’s comments (in Yellow)

• This paper is publishable upon addressing the comments provided by our peer reviewers as outlined below.

Action taken: Thanks for the comment, the manuscript was revised to align with PLOS ONE style requirements. Please consider the revised version for changes made.

Action taken: Many thanks for the comment. To address the comment, we have amended the ethical consideration subsection of method section to include information on consent as far as minors are concerned. The script added reads as “A written informed consent was obtained from the study participants to be part of the survey. Furthermore, a waiver of informed consent for participants aged 15 to 17- years was granted by RNEC. Children <18 years of age identified as being engaged in sex work, trafficked, or victim of violence, received a special post HIV test counseling and were referred for appropriate services to ensure their protection and well-being.”

3. For studies involving third-party data, we encourage authors to share any data specific to their analyses that they can legally distribute. PLOS recognizes, however, that authors may be using third-party data they do not have the rights to share. When third-party data cannot be publicly shared, authors must provide all information necessary for interested researchers to apply to gain access to the data. (https://journals.plos.org/plosone/s/data-availability#loc-acceptable-data-access-restrictions)

Action taken: Thanks for the comment, sure this manuscript involves third-party data hence cannot be availed publicly. To address the comment, we have considered describing where the data may be found and data request channels. The following script is added in the submission portal “The data underlying the results presented in the study are available for researchers who meet set criteria upon request and approval by RBC (data access requests are sent at: ‘info@rbc.gov.rw’).

Action taken: Many thanks for spotting errors in references. We considered revising citations with errors and revisions are available in the revised version of the manuscript.

Response to reviewers’ comments (in Yellow)

Introduction

The research article titled “Population size estimation of female sex workers and sexually exploited minors using privatized network sampling, Rwanda, 2023” aims to estimate the population size of female sex workers (FSW) and sexually exploited minors in Rwanda using privatized network sampling (PNS), a method that uses network information collected within a bio-behavioral survey that used respondent-driven sampling (RDS). The study collected data from 2,511 FSW across 10 study sites in Rwanda from May 8th to June 24th, 2023. Three PNS estimators were used to calculate the population size: Cross-Sample, Cross-Alter, and Cross-Network. The national-level FSW population size was estimated at 98,587 (95% CI: 82,978 – 114,196), corresponding to 2.3% of the total adult female population aged 15 years and above in Rwanda. The highest proportion of FSW was found in the City of Kigali (5.3%), followed by the West Province (2.2%), and the lowest in the North Province (1.6%). The study concludes that PNS provides fundamental information for designing, planning, and implementing programs for FSW at the provincial level in Rwanda.

Comments for improvement

• Title: the title could benefit from greater clarity, ethical consideration, and sensitivity to the populations studied. A suggested title "Estimating the Size of Vulnerable Populations: A Comprehensive Study on Female Sex Workers and Sexually Exploited Minors in Rwanda using Private Network Sampling in 2023".

Action taken: Many thanks for your suggestion, the title is modified to reflect the proposed aspects (ethical consideration and sensitivity). Kindly have a look at the revised version of the manuscript for your consideration. The title now reads as: “Estimating the Size of Hard to Sample Populations: A Comprehensive Study on Female Sex Workers and Sexually Exploited Minors in Rwanda using Private Network Sampling in 2023.”

• Abstract: The abstract provides a good overview of the background, methods, results, and conclusion of the study. However, it could be improved by:

o Providing some context and motivation for why estimating the size of FSW population is important for HIV prevention and control in Rwanda.

Action taken: Many thanks for the suggestion, much appreciated. To address this, we have considered adding a sentence describing the main benefit for the HIV prevention and control program for having the most up to date population size estimate of female sex workers in Rwanda. The sentence was added under the abstract’s introduction section and reads as: “Having population size estimates of the HIV-mostly affected population, FSW in this case provides the basis for determining the denominators to assess HIV program performance towards national and global targets of controlling the HIV epidemic among the FSW population.”

o Stating the main research question or objective of the study.

Action taken: Thanks for your comment, we have considered adding a sentence describing the main objective of this study at the end of the abstract’s introduction section. The added sentence reads as: “The aims of this study are to provide the most up to date national population size estimates (PSE) and geographical distribution of female sex workers and sexually exploited minors in Rwanda.”

o Avoiding repetition of information, such as mentioning the study period twice or the provincial distribution of FSW twice.

Action taken: Thanks for spotting the repeated information, much appreciated. To address this comment, we have considered proofreading and revising the whole abstract for clarity. All changes made are traceable in the revised version of the manuscript.

• Introduction: The introduction provides a good background and literature review on the HIV situation among FSW in Rwanda and globally, and the challenges and methods of estimating their population size. However, it could be improved by:

o Providing a clear statement of the research question or objective at the end of the introduction, and how it relates to the existing knowledge gap or research need.

Action taken: Thanks for your comment, much appreciated. To address this comment, the last paragraph of the Introduction section is revised to incorporate study main objective and how it relates to the existing research gaps in line with Rwanda HIV prevention and control program. The paragraph now reads as: “The study aims to estimate the size of FSW and sexually exploited minors operating in Rwanda using Privatized Network Sampling (PNS), capable to tap into previously unexplored FSW subgroups such as home and internet based FSWs.”

o Providing some hypotheses or expectations about the results of the study, based on the literature review or the formative assessment.

Action taken: Thanks for the comment. Re rectify this, we have considered revising the last paragraph of the introduction section to reflect anticipated scientific and programmatic contribution of the study findings based on suggested further research areas from previously conducted similar studies. This includes exploring novel approaches of population size estimation to tap into unexplored FSW’s subgroups including home and internet based FSW.

• Methods: The methods section describes the study design, procedures, sample size, data management, and statistical analysis in detail and with clarity. However, it could be improved by:

o Providing more information about the ethical considerations, such as how the informed consent was obtained, how the confidentiality and anonymity of the participants were ensured, and how the compensation was determined and delivered.

Action taken: Thanks for the comment. We have considered revising the ethical consideration section of the manuscript by incorporating all safeguard measures taken to ensure participants’ anonymity, confidentiality and a minimized potential stigmatization that might occur because of participation. These includes anonymizing participants identifiers throughout the study implementation, ensuring that participation is voluntarily, and consenting process aligns with ethical requirement. Furthermore, we have added a sentence describing at what basis compensation was determined and delivered. Kindly refer to the revised version of the manuscript for all these changes.

o Providing more information about the formative assessment, such as the number and characteristics of the FSW who participated in the group meeting and the FGD, the main findings or themes that emerged from the FA, and how they informed the design and implementation of the PNS.

Action taken: Thanks for the comment. We have considered providing further formative assessment findings and how they informed the study design and procedures, study population characteristics, and logistic component. The added content is located under study design and procedures sub-section of the method section.

o Providing more information about the statistical analysis, such as the assumptions and limitations of the PNS estimators, and the sensitivity analysis or robustness checks performed

Action taken: Many thanks for the comment, much appreciated. To address the comment, we have considered adding a paragraph under statistical analysis subsection of the methods section that describes assumptions underlining the method used and corresponding limitations. Furthermore, the paragraph shades the light on estimators’ performance assessment and selection. Please consider the revised version of the manuscript for your considerations.

• Results:

o Provide more interpretation and explanation of the results, such as the possible reasons or factors for the observed patterns, trends, or differences in the data, and the implications or significance of the results for the research question or objective.

Action taken: Thanks for suggestions, very critical. To address this comment, we considered revising the results section, mainly tables’ narratives to shade more light on findings by providing results interpretation for the observed patterns and differences in results. In addition, detailed findings discussions are found in the discussion section of the manuscript.

• Discussion: The discussion section summarizes and discusses the main findings and implications of the study and compares and contrasts them with the existing literature. However, it could be improved by:

o Providing more contextual and practical implications of the results, such as the policy and program recommendations for improving the access and quality of HIV prevention and treatment services for FSW in Rwanda, and the potential impact or benefit of the study for the HIV epidemic control and public health outcomes in the country.

Action taken: Thanks for suggestion. To address the comment, a paragraph is added at the end of discussion section, describing the implication, and potential benefits of the study results for the HIV epidemic control and public health outcomes in Rwanda. Please consider the revised version of the manuscript for additions.

o Providing more suggestions for future research, such as the limitations or gaps that need to be addressed, the questions or hypotheses that need to be tested, the methods or data that need to be improved or developed, the collaborations or partnerships that need to be established or strengthened, and the assumptions or uncertainties of the PNS method, the representativeness or quality of the data, or the generalizability or applicability of the findings.

Action taken: Thanks for the comment. To address this review comment, we considered providing a paragraph describing future research endeavors regarding the methodological limitation and study design weaknesses. Revisions might be found in the revised version of the manuscript under discussion section.

• Conclusion: The conclusion section provides a brief and concise summary of the main findings, implications, and recommendations of the study. However, it could be improved by:

o Emphasizing the key messages or takeaways of the study and provide some directions or priorities for future research, such as the areas that need further investigation, the methods or data that need refinement or innovation, or the stakeholders or partners that need engagement or support.

Action taken: Thanks for suggestion. To address the comment, the conclusion section is revised to include the key highlights from the study as well as suggestion for future studies endeavors. Please consider changes in the revised version of the manuscript.

• References:

o Check the accuracy and completeness of the citation information, such as the authors, titles, journals, volumes, issues, pages, years, DOIs.

o Check the consistency and correctness of the citation format, such as the punctuation, capitalization, italicization, or abbreviation of the elements of the citations.

Action taken: Many thanks for spotting errors in references. We considered revising citations with errors and revisions are available in the revised version of the manuscript.

---

## [Decision Letter · Decision Letter 1]

10 Sep 2024

PONE-D-23-42748R1Estimating the Size of Hard to Sample Populations: A Comprehensive Study on Female Sex Workers and Sexually Exploited Minors in Rwanda using Private Network Sampling in 2023.PLOS ONE

Dear Dr. TUYISHIME,

Thank you for submitting your manuscript to PLOS ONE. After careful consideration, we feel that it has merit but does not fully meet PLOS ONE’s publication criteria as it currently stands. Therefore, we invite you to submit a revised version of the manuscript that addresses the points raised during the review process.

We look forward to receiving your revised manuscript.

Kind regards,

Ivan Alejandro Pulido Tarquino, MSc

Academic Editor

PLOS ONE

Journal Requirements:

Additional Editor Comments:

Dear Dr. Tuyishime,

Thank you for submitting to PLOS ONE this piece of work, relevant and important for the region.

Please provide your answers and revision to reviewers.

Thank you

Kind regards

Reviewers' comments:

Reviewer's Responses to Questions

**Comments to the Author**

1. If the authors have adequately addressed your comments raised in a previous round of review and you feel that this manuscript is now acceptable for publication, you may indicate that here to bypass the “Comments to the Author” section, enter your conflict of interest statement in the “Confidential to Editor” section, and submit your "Accept" recommendation.

Reviewer #2: (No Response)

2. Is the manuscript technically sound, and do the data support the conclusions?

Reviewer #2: Yes

3. Has the statistical analysis been performed appropriately and rigorously? 

Reviewer #2: Yes

4. Have the authors made all data underlying the findings in their manuscript fully available?

Reviewer #2: Yes

5. Is the manuscript presented in an intelligible fashion and written in standard English?

Reviewer #2: Yes

6. Review Comments to the Author

Reviewer #2: Authors stated: 'The survey was reviewed and approved by the Rwanda National Ethics Committee (RNEC).' Thus protocol ID is highly needed.

7. PLOS authors have the option to publish the peer review history of their article (what does this mean? ). If published, this will include your full peer review and any attached files.

**Do you want your identity to be public for this peer review?** For information about this choice, including consent withdrawal, please see our Privacy Policy .

Reviewer #2: No

---

## [Author Response · Author response to Decision Letter 2]

30 Oct 2024

Responses to ACADEMIC EDITOR’s comments (in Yellow)

1. Journal Requirements:

Please review your reference list to ensure that it is complete and correct. If you have cited papers that have been retracted, please include the rationale for doing so in the manuscript text or remove these references and replace them with relevant current references. Any changes to the reference list should be mentioned in the rebuttal letter that accompanies your revised manuscript. If you need to cite a retracted article, indicate the article’s retracted status in the References list and also include a citation and full reference for the retraction notice.

Action taken: Thanks for the comment, the manuscript was revised to ensure that the list of all cited references is complete, and no retracted paper are referenced.

2. Reviewer #2:

Authors stated: 'The survey was reviewed and approved by the Rwanda National Ethics Committee (RNEC).' Thus, protocol ID is highly needed.

Action taken: Thanks for the comment, the protocol ID number of the protocol covering the survey is added under “Ethical consideration” section (page #9) as it appears on the approval letter secured from Rwanda National Ethics Committee (ID: IRB 00001497 of IORG0001100.

---

## [Decision Letter · Decision Letter 2]

13 Dec 2024

PONE-D-23-42748R2Estimating the Size of Hard to Sample Populations: A Comprehensive Study on Female Sex Workers and Sexually Exploited Minors in Rwanda using Private Network Sampling in 2023.PLOS ONE

Dear Dr. Elysee Tuyishime,

Thank you for submitting your manuscript to PLOS ONE. After careful consideration, we feel that it has merit but does not fully meet PLOS ONE’s publication criteria as it currently stands. Therefore, we invite you to submit a revised version of the manuscript that addresses the points raised during the review process. Please submit your revised manuscript by Jan 27 2025 11:59PM. If you will need more time than this to complete your revisions, please reply to this message or contact the journal office at plosone@plos.org . Please include the following items when submitting your revised manuscript:

We look forward to receiving your revised manuscript.

Kind regards,

Ivan Alejandro Pulido Tarquino, MSc

Academic Editor

PLOS ONE

Journal Requirements:

Additional Editor Comments (if provided):

Dear Dr. Elysee Tuyishime,

Thank you for submitting your manuscript, "Estimating the Size of Hard to Sample Populations: A Comprehensive Study on Female Sex Workers and Sexually Exploited Minors in Rwanda using Private Network Sampling in 2023", to PLOS ONE.

The review process has now been completed, and the reviewers have provided their feedback and suggestions for improvement. I kindly request that you address the comments and revise your manuscript accordingly. Once the revisions are complete, please re-submit the updated manuscript for further evaluation.

Should you have any questions or need clarification on the reviewers' comments, feel free to reach out. We are here to support you through the revision process.

We look forward to receiving your revised manuscript.

Best regards,

Dr. Ivan Alejandro Pulido Tarquino,

Academic editor

Reviewers' comments:

Reviewer's Responses to Questions

**Comments to the Author**

1. If the authors have adequately addressed your comments raised in a previous round of review and you feel that this manuscript is now acceptable for publication, you may indicate that here to bypass the “Comments to the Author” section, enter your conflict of interest statement in the “Confidential to Editor” section, and submit your "Accept" recommendation.

Reviewer #3: (No Response)

Reviewer #4: (No Response)

2. Is the manuscript technically sound, and do the data support the conclusions?

Reviewer #3: Yes

Reviewer #4: Yes

3. Has the statistical analysis been performed appropriately and rigorously? 

Reviewer #3: Yes

Reviewer #4: Yes

4. Have the authors made all data underlying the findings in their manuscript fully available?

Reviewer #3: Yes

Reviewer #4: Yes

5. Is the manuscript presented in an intelligible fashion and written in standard English?

Reviewer #3: Yes

Reviewer #4: Yes

6. Review Comments to the Author

Reviewer #3: While detailed, the methodology had gaps: the inclusion criteria lacked clarity on validating “self-identified FSWs”. The formative assessment involved a limited, potentially unrepresentative sample. Recruitment processes risked bias, and logistical challenges were inadequately addressed. Sample size justification, and strategies for isolated FSWs need strengthening. Data validation processes for duplicate IDs and network anomalies were insufficiently detailed, and adjustments for marginalized subgroups (e.g., those without cell phones) were unclear. Provincial disparities were overlooked, warranting more granular analysis.

The discussion highlights the use of PNS to estimate the population size of FSWs in Rwanda, revealing a larger estimate compared to previous studies due to improved capture of non-venue-based FSWs. However, it inadequately analyzes differences between PNS and prior methods, underexplores subgroup insights and overrelies on static HIV program data for provincial estimates. Limitations like SIM card access and PNS assumptions are acknowledged but not critically quantified. Recommendations for future research lack depth, focusing narrowly on small-area estimates. The section could benefit from clearer organization, critical evaluation, and specific programmatic implications.

Reviewer #4: Please see the attachment that contains specific details on the manuscript from the perspective of the third reviewer.

7. PLOS authors have the option to publish the peer review history of their article (what does this mean? ). If published, this will include your full peer review and any attached files.

**Do you want your identity to be public for this peer review?** For information about this choice, including consent withdrawal, please see our Privacy Policy .

Reviewer #3: No

Reviewer #4: **Yes: ** Sarah Moreheart

---

## [Author Response · Author response to Decision Letter 3]

28 May 2025

Dear Handling editor, I am kindly submitting the revised version of the manuscript that addresses the points raised during the review process.

I have attached the following:

- A rebuttal letter that responds to each point raised by the academic editor and reviewer(s), labeled with 'Response to Reviewers'

- A marked-up copy of the manuscript that highlights changes made to the original version, labeled with 'Revised Manuscript with Track Changes'

- An unmarked version of the revised paper without tracked changes, labeled with 'Manuscript'.

Many thanks for great comments, much appreciated.

Regards.

---

## [Decision Letter · Decision Letter 3]

22 Jul 2025

Estimating the Size of Hard to Sample Populations: A Comprehensive Study on Female Sex Workers and Sexually Exploited Minors in Rwanda using Private Network Sampling in 2023.

PONE-D-23-42748R3

Dear Dr. Elisee Tuyishime,

We’re pleased to inform you that your manuscript has been judged scientifically suitable for publication and will be formally accepted for publication once it meets all outstanding technical requirements.

Kind regards,

Ivan Alejandro Pulido Tarquino, MSc

Academic Editor

PLOS ONE

Additional Editor Comments (optional):

Dear Dr. Tuyishime,

We are pleased to inform you that your manuscript entitled “Estimating the Size of Hard to Sample Populations: A Comprehensive Study on Female Sex Workers and Sexually Exploited Minors in Rwanda using Private Network Sampling in 2023.”, submitted to PLOS ONE, has been reviewed and deemed idoneous for publication.

We would like to commend you and your co-authors for the substantial improvements made throughout the revision process. The final version of the manuscript demonstrates a significant enhancement in clarity, scientific rigor, and overall quality. We particularly appreciate the thorough and thoughtful manner in which you addressed each of the reviewers' comments, your responses clearly contributed to strengthening the manuscript and reflect a high level of academic engagement.

The editorial team will now proceed with the final steps toward publication. Should any minor revisions or administrative formalities be needed, our office will contact you shortly.

Congratulations on this achievement. We look forward to publishing your work, and thank you for your valuable contribution to the field.

Kind regards,

Ivan Alejandro Pulido Tarquino

Reviewers' comments:

Reviewer's Responses to Questions

**Comments to the Author**

1. If the authors have adequately addressed your comments raised in a previous round of review and you feel that this manuscript is now acceptable for publication, you may indicate that here to bypass the “Comments to the Author” section, enter your conflict of interest statement in the “Confidential to Editor” section, and submit your "Accept" recommendation.

Reviewer #4: All comments have been addressed

2. Is the manuscript technically sound, and do the data support the conclusions?

Reviewer #4: Yes

3. Has the statistical analysis been performed appropriately and rigorously? 

Reviewer #4: Yes

4. Have the authors made all data underlying the findings in their manuscript fully available?

Reviewer #4: Yes

5. Is the manuscript presented in an intelligible fashion and written in standard English?

Reviewer #4: Yes

6. Review Comments to the Author

Reviewer #4: Thank you for the opportunity to review this article. I am satisfied that all the suggestions and revisions have been addressed.

7. PLOS authors have the option to publish the peer review history of their article (what does this mean? ). If published, this will include your full peer review and any attached files.

**Do you want your identity to be public for this peer review?** For information about this choice, including consent withdrawal, please see our Privacy Policy .

Reviewer #4: **Yes: ** Sarah Moreheart

---

## [Editor Report · Acceptance letter]

PONE-D-23-42748R3

PLOS ONE

Dear Dr. TUYISHIME,

I'm pleased to inform you that your manuscript has been deemed suitable for publication in PLOS ONE. Congratulations! Your manuscript is now being handed over to our production team.

Kind regards,

on behalf of

Dr. Ivan Alejandro Pulido Tarquino

Academic Editor

PLOS ONE